# Using attribution theory to explore the reasons adults with hearing loss do not use their hearing aids

**Caitlyn R. Ritter**[1]**, Brittan A. Barker**[1]\***, Kristina M. Scharp**[2]

**1** Department of Communicative Disorders and Deaf Education, Utah State University, Logan, Utah, United States of America, **2** Department of Communication, University of Washington, Seattle, Washington, United States of America

\* brittan.barker@usu.edu

**Data Availability Statement:** All relevant data are within the manuscript and its Supporting Information files.

**Funding:** The author(s) received no specific funding for this work.

## Abstract

Hearing aids are an effective treatment for individuals with hearing loss that have been shown to dampen (and sometime ameliorate) the negative effects of hearing loss. Despite the devices' efficacy, many reject hearing aids as a form of treatment. In the present qualitative study, we explored the reasons for hearing aid non-use in the United States that emerged from the stories of adults with hearing loss who do not to utilize hearing aids. We specifically used thematic analysis in concert with an attribution theory framework to identify and analyze recurring themes and reasons throughout these individuals' narratives. A total of nine themes describing reasons of hearing aid non-use emerged. Four reasons were internally motivated: (1) non-necessity, (2) stigmatization, (3) lack of integration into daily living, and (4) unreadiness due to lack of education; five reasons were externally motivated: (5) discomfort, (6) financial setback, (7) burden, (8) professional distrust, and (9) priority setting. These findings contribute to the field of hearing healthcare by providing professionals with insight into reasons that people across the provided when recounting their experiences following the diagnosis of hearing loss, prescription for hearing aids, and their hearing aid non-use. These findings are an important step toward the development of more effective, person-centered hearing healthcare that can best address these individuals' concerns and expectations surrounding hearing loss and hearing aids.

## Introduction

Hearing loss (HL) is increasingly prevalent and affects people regardless of age, gender, or race, with some data showing that 48.1 million individuals 12 years and older living in the United States (U.S.)—about 20% of the entire population—have HL [1]. Further, of those diagnosed with HL, the majority of them do not use hearing aids (HAs) [2–4]. Without listening-device intervention, there are a number of negative consequences that can stem from HL, which range from psychological issues (e.g., negative effects on quality of life and well-being) [5] to accelerated cognitive decline [6]. The negative ramifications coupled with reluctance for

**Competing interests:** The authors have declared that no competing interests exist.

HA usage is concerning to hearing healthcare providers, considering HA acceptance and use can mitigate these negative consequences [7–10]. Despite this concern, researchers have yet to document in-depth, explict reasons why people with HL choose not to wear their HAs [3]. Consequently, the purpose of this study was to gather stories from people with HL and illuminate the attributions—both internal (resulting from personal factors/characteristics) and external (resulting from a situation/context that a person is in)—they make for not wearing their HAs. Better understanding the reasons people do not to wear their HAs can enable hearing healthcare providers to provide improved, person-centered care [11, 12] that could minimize patients' barriers to HA use and subsequently foster HA acceptance and consistent usage.

## Establishing the importance of HA use

Despite the fact that HAs' efficacy is unquestionable [7–10], individuals with HL do not often utilize listening technology to manage their HL [2–4]. A national survey conducted by Lin, Thorpe, Gordon-Salant, and Ferrucci [13] across the U.S. revealed that 63.1% of individuals 70 years or older had HL (n = 717). Furthermore, there were notable differences across individuals when looking at HA uptake rates depending on degree of HL: 76.6% percent of adults with severe HL and 40% percent of adults with moderate HL used HAs, while only 3.4% of adults with mild HL used HAs. These HA uptake statistics are relatively low given the benefits that HAs provide, but the nearly 97% of adults with mild HL reported to not use HAs is especially concerning given that research suggests HA use is remarkably beneficial for these individuals [10].

## The varying experience of HA users

Existing research involving people who use their HAs, albeit sparse, provides important insight and serves as additional rationale to better understand those who choose not to use their HAs. For example, Bennett, Laplante-Lévesque, Meyer, and Eikelboom [14] found that the HA owners and hearing healthcare providers had similar perceptions about a variety of common problems associated with HAs (e.g., management; sound quality and performance; negative feelings, thoughts and behaviors). What is important to note is that when asked to determine how detrimental these problems were to consistent HA use, *providers* perceived that these problems were more severe than the *patients* did. These findings are valuable because they not only highlight the disconnect between providers' perspectives and patients' experiences but also allude to the possibility that perceived problems are not inherently the reasons why those with HL do not wear their HAs.

One potential reason for this disconnect is that not all people with HL have the same experience with HAs. Barker, Scharp, Long, and Ritter [15] recently employed a narrative approach to uncover the identities of adults with HL who reported consistently using their HAs. Specifically, in their qualitative study, they used thematic narrative analysis (TNA) [16] and analyzed narratives from HA users (N = 30). Their TNA illuminated five distinct identities that HA users constructed in response to their HL diagnosis and HA adoption: (1) the satisfied user, (2) the overcomer, (3) the dispassionate user, (4) the resigned user, and (5) the griever. Thus, these findings not only revealed that HA users vary in their experiences with HAs, but also that the reasons for HA non-use might be more complex and extend beyond issues of general satisfaction and perceived device-centric barriers as previously noted [3, 17]. In other words, Barker and colleagues' [15] research points to the importance of understanding why people choose not to wear their HAs, instead of assuming a dichotomous experience between people who are consistent users versus people who are not. Given the mismatch between provider and patient perceptions and the complexity surrounding HL diagnosis and HA use, direct

accounts from HA non-users could provide an imperative perspective that could serve to help hearing healthcare providers understand why the HAs they prescribe are not being used.

## The experience of HA non-use

There is a lack of empirical data focused on HA non-use, which could be contributing to professionals' lack of understanding regarding poor HA use and adherence to treatment. This is clearly problematic considering HA users likely do not require the same professional, emotional, and educational support from hearing healthcare providers as those who dismiss the recommendation and prescription for HAs [18]. As evidenced in their qualitative study, Linssen, Joore, Minten, van Leeuwen, and Anteunis [19] explored older, Dutch individuals' beliefs and feelings toward their HL and HA non-use through semi-structured interviews. Using narrative analysis [16], they revealed that 11 adults with HL perceived: (1) their HL as a handicap, (2) both internal and external factors were responsible for their HA non-use, and (3) their significant other's attitudes contributed to their HA non-use. Although these findings offer indispensable, foundational insight into individuals' perceptions of their HL and HAs, the generalizability of their results might be limited and in many ways lack specificity. For example, the fact that the participants' HAs were procured within the Netherlands' universal healthcare system managed by the government and supplemented by private insurers might not reflect the experiences of adults' residing in other countries that lack universal healthcare and HA coverage (e.g., the U.S.). Another example of a potentially limiting factor of Linssen and colleagues' [19] study is the semi-structured interview format [20] they employed. Although semi-structured interviews provide some flexibility when it comes to gathering information from research participants, they also come with the risk that individuals will adapt their answers to what the researcher wants to hear (as guided by the questions or prompts). Alternatively, a narrative approach [21]—in concert with an attribution theory framework—would allow researchers to thoroughly explore individuals' reasons and beliefs for their HA non-use via the participants own stories highlighting what they prioritize.

## Personal narratives and an attribution theory framework

People regularly tell stories (i.e. narratives) to communicate. We tell stories both to ourselves and others as a way to make sense of our thoughts/experiences and to organzie and understand our lives' happenings [21, 22]. Herman [23] defines a narrative as "a basic human strategy for coming to terms with time, process, and change." Research shows that telling stories influcene one's well-being, self-esteem, and overall self concept [24]. In addition to telling stories, it has been shown that humans often naturally work to find causal connections between events in their lives and the causes of their own (and others') behaviors. This is known as attribution theory [25]. Attribution theory specifically attempts to explain individuals' reasons for their behaviors and the degree to which these reasons are internally or externally motivated. Internal attributions suggest that life happenings (and/or one's outlook) are a result of a person's own attributes and are within their control. Alternatively, external attributions drive behaviors that can be attributed to environmental factors outside of the individual's control. Given that both narratives and attribution theory center on people's internal dialogues and thought processes, it is not surprising to learn that attributional concepts are often used in definitions of narrative [26]. In fact, one can easily aruge, as Robinson and Hawpe [27] did, that "Narrative thinking, is therefore a type of causal thinking."

Recall, research demonstrated that HA use benefits people with HL audiologically, psychologically, socially, and physiologically [7–10]. Thus, individuals who are prescribed HAs but do not use them could be perceived as engaging in counterintuitive behavior. Although

previous research showed there are indeed reasons people with HL choose not to use HAs (see [3] for review), to date no one has gathered stories from these individuals to explore their explicit reasons (and underlying beliefs) for HA non-use. Attribution theory—in combination with personal narratives—provides a framework in which we can begin to identify and explore important non-audiological factors that contribute to HA use [17]. This knowledge could subsequently inform person-centered hearing healthcare [11, 28] and improve providers' abilities to engage in better and more efficient intervention that directly addresses patients' perceived barriers, whether internal or externally motivated, to hearing device uptake. For example, consider barriers that are externally driven, such as the potential financial burden of HAs or other listening devices. A possible solution to overcome such a financial burden might include improving a hearing healthcare professional's knowledge and implementation of federal or private funding programs for hearing devices available to the patient at little to no cost.

We propose that gathering unrestricted personal accounts of HA non-users and exploring the reasons for non-use that emerge from their stories is both clinically and theoretically important. This study is a first step toward implementing improved person-centered audiological care [11] grounded in the International Classification of Functioning, Disability and Health framework (ICF) [29]. By evaluating a person holistically and working with them to determine their level of functioning in the world, their hearing healthcare can be better tailored to fit their needs of daily living; thus, having the potential to result in improved patient HA compliance and satisfaction [28]. By identifying attributions in people's narratives, we are able to more appropriately explore the variety of reasons that contribute to HA non-use and foundational knowledge about a distinctive patient population. Thus, we asked the following research question: What reasons do adults with HL, who are prescribed HAs, provide for not using HAs?

## Materials and methods

The Utah State University institutional review board approved this study; IRB#8063. All participants provided written consent prior to their participation.

### Study design

In this study, we engaged in qualitative analysis of participants' narrative transcripts. Specifically, narrative interviews allow participants to share their story their way and are the most open-ended type of interview [21]. Allowing participants to share their stories ensures that they are discussing the issues most important to them.

### Sampling

People were eligible for participation in the study if they (1) were at least 18-years-old; (2) communicated comfortably using spoken English; (3) reported typically functioning cognition, with no co-occurring speech or language disorders; (4) were professionally diagnosed with HL; (5) were prescribed HAs by a hearing healthcare provider, and (6) self-identified as a HA non-user. We aimed recruit a heterogeneous sample with regard to age, gender, degree of HL, and residence. We thus recruited participants for the study in the following manners: (1) distributed letters to patients at the university's audiology clinic, (2) posted fliers in local, public spaces, (3) spoke to classrooms of individuals participating in the university's *Summer Citizens* program, (4) posted recruitment information on social media and lab websites, and in local newsletters, and (5) word of mouth. Potential participants were instructed to contact the researchers if they wanted to participant and met the inclusion criteria. After they reached out, a researcher sent an email to the individual with a survey link (see *Phase I* below for more details).

## Participants

Twenty adults (n = 20), who self-identified as HA non-users, participated in the study. All participants were diagnosed with varying degrees of HL by a hearing healthcare provider who recommended that HAs would be beneficial. These individuals ranged in age from 27–91 years (M = 65.6 years; SD = 15.5). All participants resided in the U.S. and primarily identified as White. Participants in the study met the aforementioned inclusion criteria. It is interesting to note that although all eligible participants perceived themselves to be "non-users", they varied in their response to the question "How often do you wear your hearing aids?" Table 1 displays the participants' demographic information. Note that all of the participants were included in the final data set, regardless of quantitative measures/reports of HA use, because they self-identified as HA non-users. As narrative researchers, we were interested in the participants' *perceptions* of themselves and the attributions that result from said perceptions [30].

## Procedure

Procedures for this study were reviewed and approved by the university's Institutional Review Board.

**Phase I.** When an individual reached out to the lab and expressed interest in participating, they were sent a link to the aforementioned *Qualtrics* [31] questionnaire via email. The potential participant then completed the questionnaire to confirm they met the study's inclusion

**Table 1. Participant demographic information and HA usage.**

| Participant | Age | Gender | Ethnicity | PTA$^3$ (dB HL) | HA utilized? | HA usage |
|---|---|---|---|---|---|---|
| Maggie, #1 | 62 | F | White | 28 | no | N/A |
| Chelsea, #2 | 27 | F | White | - | no | N/A |
| Ron, #3 | 53 | M | White | 15 | no | N/A |
| Donna, #4 | 63 | F | White | 25 | yes | never during waking hours |
| Patrick, #5 | 73 | M | White | - | yes | sometimes during waking hours |
| Doug, #6 | 71 | M | White | 18 | yes | sometimes during waking hours |
| Karly, #7 | 73 | F | White | 28 | no | N/A |
| Susan, #8 | 79 | F | White | 47 | no | N/A |
| Anthony, #9 | 88 | M | White, Italian | - | yes | never during waking hours |
| Steve, #10 | 81 | M | White | 18 | yes | sometimes during waking hours |
| Adam, #11 | 91 | M | White | 28 | no | N/A |
| Violet, #12 | 63 | F | White | 38 | no | N/A |
| Anna, #13 | 75 | F | White | 23 | no | N/A |
| Paul, #14 | 67 | M | did not report | 25 | yes | during most waking hours |
| Mike, #15 | 48 | M | White | - | yes | sometimes during waking hours |
| Carmen, #16 | 69 | F | White | - | no | N/A |
| Ryan, #17 | 67 | M | White | 50 | no | N/A |
| Phillip, #18 | 51 | M | White | - | yes | during all waking hours |
| Kennedy, #19 | 42 | F | White | 33 | yes | during about half of my waking hours |
| Meghan, #20 | 69 | F | White | 20 | yes | during most waking hours |

Participant = participant's pseudonym and their interview number for the present study; age is reported in years; F = female; M = male; PTA$^3$ is the pure-tone average of 500 Hz, 1000 Hz, and 2000 Hz for the better ear; HA utilized is whether the participant utilized HAs after the initial prescription; HA usage is self-reported by the participant via the initial demographic survey asking: "How often do you/did you use your hearing aids?" (i.e. *during all waking hours*, *during most waking hours*, *during about half of my waking hours*, *sometimes during waking hours*, *never during waking hours*)."-" = data not available. N/A = Not applicable, as the patient never utilized HAs after their diagnosis of HL and prescription for HAs; the patient only utilized HAs during a trial period.

criteria. This questionnaire included an IRB-approved letter of informed consent, a Health Insurance Portability and Accountability Act (HIPAA) authorization form, and a series of 10 demographic questions. After each individual completed the questionnaire, a researcher then scheduled an interview with them.

**Phase II.** The first author, a female researcher with extensive training in clinical audiology and narrative interviewing, conducted the majority of interviews. A second female researcher, also with extensive training in clinical audiology and narrative interviewing, conducted interviews when scheduled warranted her help (n = 8). Both interviewers followed the same interview script without deviation (see S1 Appendix). Interviews were conducted face-to-face (n = 6) in the second author's lab or via telephone (n = 14). At the beginning of the interview, the researcher briefly introduced herself as a student researcher working on her clinical research project to earn her doctorate of audiology degree. She then confirmed that the participant gave their informed consent via the email questionnaire. All interviews were recorded using a digital-audio recorder paired with a microphone. Only the researcher and participant were present during the interview, with the exception of RP8's face-to-face interview when her spouse was present. Following the interview, the participant was compensated for their time, either with $10 cash or a $10 electronic gift card. The mean interview time was 20 minutes 10 seconds, with a range of 10 minutes 7 seconds to 36 minutes 12 seconds. All participants completed the interview and remained in the study. No repeat interviews were conducted.

**Phase III.** After data collection was complete, the interviewer uploaded the audio files from the digital recorder to a personal computer. Three research assistants experienced with narrative transcription then used a word processing program on a personal computer running *Express Scribe Transcription Software Pro v 6.10* and *Dragon Dictation v15* [32] paired with a transcription foot pedal and circumaural headphones to transcribe the recorded interviews. During the transcription process, any proper names and places mentioned by the participants were replaced with pseudonyms. The 20 completed transcriptions yielded a total of 97 pages of single-spaced text. None of the transcripts were returned to the participants for comment or correction. Finally, we contacted each participant's hearing healthcare provider to confirm their diagnosis of HL and (when applicable) we collected specific information about the participant's listening devices. These healthcare data can be found in Table 1.

## Data analysis

The authors coded the data and employed thematic analysis (TA) [16] to identify commonly expressed themes within the narrative corpus. Given that the aim of this study was to uncover the reasons adults with HL who do not to use their HAs we took a theoretically driven approached and employed the Attribution Theory framework [25]. The unit of analysis in the present study was the utterances within the transcribed dialogue from a single individual's interview. For the purpose of this study, we conceptualize an utterance as a turn in talk. We followed a TA procedure to assign reasons for HA non-use to each of the 20 participants based on the unit of analysis. To assign the reasons, we engaged in the following general steps of TA: 1) became familiar with the data, 2) generated initial codes, 3) identified the themes coalesced from codes, 4) reviewed themes, 5) defined and named the themes, and 6) located compelling exemplars.

## Verification procedures

To verify our qualitative findings, we employed the following four procedures common in vigorous qualitative research (see TA analysis verifications): (1) referential adequacy, (2) peer debriefing, (3) audit trail, and (4) exemplar identification [33]. Referential adequacy was accomplished by first splitting the data in half at Interview #10. Analysis of the first 10

interviews was independently and thoroughly conducted by the first and second authors. During peer debriefing, we discussed differences in our findings and came to a final agreement across all defined themes and codes amongst the first and second authors and then amongst the first, second, and third authors. We defined nine themes after the analysis of the first 10 interviews. We then analyzed the remainder of the data, which included 10 more interviews; no additional themes emerged during analysis of these remaining narratives. Saturation was reached at Interview #11 [34]. Throughout the analysis, we kept detailed electronic notes in *NVivo* [34] to track the decisions we made when identifying codes and reasons for non-use (i.e. audit trail). Lastly, we found exemplars from the participants' narratives to best illustrate each reason of HA non-use that emerged (i.e. exemplar identification). Participants did not provide feedback on the findings.

## Results

Analysis revealed internally and externally motivated reasons of HA non-use. A total of nine themes emerged within these supra-themes: four reasons for non-use that were internally motivated and five reasons that are externally motivated. Internally motivated reasons of non-use include: (1) non-necessity, (2) stigmatization, (3) lack of integration into daily living, and (4) unreadiness due to lack of education. Externally motivated reasons of non-use include: (5) discomfort, (6) financial setback, (7) burden, (8) professional distrust, and (9) priority setting. See Table 2 for brief descriptions of the themes and reasons that arose from the analysis of the narratives. Note, some participants recounted multiple reasons within their narratives (e.g., Ron, Interview #3) while some participants' narratives revealed a single reason for non-use.

### Internally motivated reasons for HA non-use

**Non-necessity.** HAs are a non-necessity was the most prevalent internally motivated reason for HA non-use within the present findings. Overall, people with narratives that suggested HAs are a non-necessity revealed that they could hear adequately without the HAs and the listening devices did not provide any subjective benefit; therefore, it was not necessary to use them. Both Steve and Phillip's narratives in particular revealed that HAs are unnecessary devices due to their lack of perceived benefit from the devices. Steve mentioned: "So, I bought them, and I wore them for a while, and they seem to work okay. But I just slowly stopped wearing them. They—and I've tried a few times and they don't seem to make any difference and so. . ." (Steve, Interview #10) Phillip similarly mentioned, "During my trial phase of using the hearing aids, um—I did not seem to recognize any kind of true benefit of hearing something that I didn't hear before. I—I couldn't—I could not distinguish that—that there were less, 'What did you says?'" (Phillip, Interview #18)

**Stigmatization.** Narratives describing ways that HAs are stigmatized introduced the reality that individuals with HL and HAs remain stigmatized in the U.S. [35]. For example, ageism [36] has been at the center of long-standing HA stigmatization, with the bias that only individuals who are older are affected by HL and this perspective was noted within our narratives. One young participant who unexpectedly found out she had HL said, ". . .then he asked me if I had ever considered getting hearing aids and I told him, 'No!' because I was only 20 years old at the time." (Chelsea, Interview #2) While aging and its physiological changes are concerns for some, vanity also surfaced as a reason for HA non-use. Donna spoke of how her audiologist addressed her concerns about vanity: "It's a little embarrassing to have to wear 'em. But the way Dr. Johnson fits it—he got some that matched my hair color and you really couldn't see 'em." (Donna, Interview #4)

**Table 2. Narrative themes and reasons for hearing aid non-use.**

| causal factor | themes + contributing reasons | description of themes + contributing reasons |
|---|---|---|
| Internal | **Non-necessity** | **These people do not believe that they benefit from HAs.** |
| | *I can manage without.* | *RP believes they can hear fine without HAs.* |
| | *I don't think HAs make a difference.* | *RP believes hearing aids do not provide any benefit.* |
| | **Stigmatization** | **These people associate hearing aids with old age, financial difficulties, and unattractive individuals.** |
| | *I'm embarrassed to wear HAs.* | *RP believes it is embarrassing to be seen with HAs on.* |
| | *I'm too young for HAs.* | *RP believes they are too young for hearing aids* |
| | *I'm too ashamed to ask for financial help.* | *RP is anxious about the process of securing HAs (e.g., going to audiologist or vocational rehabilitation counselor)* |
| | **Lack of integration into daily living** | **These individuals cannot find a way to integrate HAs into their daily lives.** |
| | *I don't want to add more stuff into my life.* | *RP does not want to add another process or thing into their life.* |
| | *I forget to use my HAs.* | *RP forgets to use HAs.* |
| | *I don't like wearing HAs.* | *RP does not like wearing HAs.* |
| | **Unreadiness due to lack of education** | **These individuals do not feel educated enough to be ready and follow through with HAs.** |
| | *I want more education before I feel ready.* | *RP believes they need more education prior to following through with HAs.* |
| External | **Discomfort** | **These people report that the HAs themselves are uncomfortable and/or the amplification causes discomfort.** |
| | *HAs caused an allergic reaction.* | *RP experiences an allergic reaction after wearing HAs.* |
| | *Amplified sounds are bothersome.* | *RP is bothered by the amplified sounds that the hearing aid provides.* |
| | *HAs are too itchy.* | *RP experiences physical irritation from the HA earmolds/domes.* |
| | *HAs exacerbate tinnitus.* | *RP believes HAs make their tinnitus worse.* |
| | *HAs are uncomfortable* | *RP believes the HAs to be uncomfortable* |
| | **Financial setback** | **These people believe HAs are too expensive.** |
| | *HAs are too expensive.* | *RP believes cost of HAs is a barrier.* |
| | **Burden** | **These individuals perceive HAs as burdensome.** |
| | *HAs require lots of fidgeting/effort.* | *RP believes HAs require too much fussing with.* |
| | **Professional distrust** | **These individuals do not trust the professionals who diagnosed the hearing loss or prescribed the HAs.** |
| | *I don't trust the HA recommendation.* | *RP does not trust the recommendation that they utilize HAs.* |
| | **Priority setting** | **These individuals do not prioritize HAs at this time.** |
| | *Other health conditions take priority* | *RP has other health conditions that take precedent over HA use.* |
| | *Other family members' needs take priority* | *RP's family members have needs that take precedent over HA use.* |

**Lack of integration into daily living.** These individuals' narratives suggested that the reason for their HA non-use is they were unable to successfully integrate listening devices into their daily lives. While the underlying reasons varied across individuals—whether it was that they believed improvements in audibility to be too complicated of a process to add into their life or they simply forgot to wear the devices—a barrier existed in the successful integration of HAs into their activities of daily living. Doug recounted his experience with HAs, noting: "So I take my hearing aids wherever we go, and they have a nice vacation and they sit in a little tub. And uh, I have a tendency to forget to, uh, wear them." (Doug, Interview #6) While Doug forgot to wear his HAs, Kennedy disliked wearing her HAs so much that she was not able to integrate them into her daily life:

I don't know if it's just because they weren't 100% custom fit, I don't know if it was just cause, like, I couldn't get 'em in right. 'Cause like I said, they were the kind that went behind my ear and then they had the little thingy, the little tube thingy, but then at the end of it

they had that little piece of, I don't like—I don't know technical terms, but they had that little piece of plastic that you kinda had to situate in your ear anyways, and it made everything louder. Um—like—and I don't like 'em. (Kennedy, Interview #19)

**Unreadiness due to lack of education.**   These narratives suggested resistance to HA use because the individuals felt uneducated about hearing devices, thus they were not ready to follow through with the recommendation and purchase of HAs. Maggie noted: "Or—you know —I just—at this point—don't—still don't want them but—um—I don't know really enough about 'em to be so confident in saying I'm not ready." (Maggie, Interview #1)

## Externally motivated reasons for HA non-use

**Discomfort.**   A variety of reasons for HA non-use emerged from individuals' narratives centered on the perception that HAs are uncomfortable. This was the most prevalent externally motivated reason for HA non-use within the present findings. The specific reasons ranged from general aches, allergic reactions, to uncomfortable loudness—thus these people perceived their HAs as too uncomfortable to utilize. Anthony, who recounted wearing his HAs consistently for only a few years, stated:

. . .as I got more in deficit of hearing, noise—surrounding noise—bothered me twice as much or three times as much. So, if there was a noise fifty feet away from me, it could bother me. As this began to get worse and worse and worse, I noticed that when we—when she—was preparing dinner or lunch or whatever, and I was sitting there reading the paper or listening to the news or whatever, it [the noise] began to bother me so much that at first just took them [HAs] out while she was cooking. And then after that, I just kind of gave up on them and I only—if something that was really important for me to hear, I would try to wear them and as soon as that situation is was over with, I didn't use them again. (Anthony, Interview #9)

While Anthony suggested his discomfort was a result of the amplified sounds, Donna deemed the HA domes to be a physical discomfort resulting in her non-use:

I was allergic to those little domes—it—and I got an ear infection. So—I had to not wear the hearing aids for a while, but I started up again about a—a month later because we had special [ear]molds made with this—um—hypoallergenic stuff. Well—it worked okay. I wasn't allergic to that, but the earmolds hurt my ears and they just itched all the time. (Donna, Interview #4)

**Financial setback.**   Many individuals' narratives revealed cost as a reason for HA non-use. For example, Patrick simply stated: "Considering the price of it [HAs], I didn't think I was getting that much, that they were giving me that much benefit, it's pretty simple." (Patrick, Interview #5) However, for Ron, his HA non-use stemmed from the belief that the costs associated with HAs would require funds to be taken from financial reserves that would otherwise go towards his family's expenses (e.g., mortgage payment). "And my biggest fear was that I would be costing—taking money out of my family's pocket for something that I could just somehow work my way around, and get through life, and just work harder, and try to do better at getting into situations where I could hear better. (Ron, Interview #3).

**Burden.**   Another externally motivated reason for HA non-use was grounded in the perception that HAs are burdensome, i.e. they require altering and "fussing with". Maggie explicitly expressed concerns regarding her belief that HAs are an inconvenience as noted from what she observes her peers with HL experience, adding:

> I just haven't pursued it [getting HAs] mostly because I find people who wear hearing aids are constantly—I'll use the word fussing with them. They're just—they [HAs] always seem to be a fussy thing. It's, it's not something like—an analogy might be contact lenses. You put them in in the morning and you go on with the day and you don't have to stop and—well, you might have to polish 'em or something but it's, it's generally not a big deal. It just seems like a number of people that I have seen with hearing aids are like, 'Oh, just a moment. Let me make an adjustment here. I didn't quite get that,' or, uh, you know, 'I have to make an adjustment when I'm in the music venue on my devices so I can hear it differently,' or in the bar or wherever we go—there's some adjustment to be made. And, and that is my major deterrent. (Maggie, Interview #1)

**Professional distrust.**   Some individuals' narratives revealed that they distrusted the hearing healthcare professionals who diagnosed their HL and recommended HAs. They used verbiage consistent with feeling surprised, dissatisfied, and unhappy while narrating their experiences. Thus, because these people did not trust their providers' diagnoses and/or recommendations they subsequently did not use HAs. Susan reflected on her experience:

> I'm not—wasn't—happy with the testing. I thought that there should have been more. But I was diagnosed with needing hearing aids and he said, 'Both, two. Both ears'. And so, when I left there, I just decided that I didn't know whether I should believe it. I really wanted to see an ear doctor. I thought that that would give me a better reading of my problem. (Susan, Interview #8)

While Susan felt dissatisfied, Carmen was surprised to receive an unexpected hearing aid recommendation, followed by the professional's assumptions about her willingness to utilize the listening devices:

> So, we went into a booth—I went into a booth—took the hearing test and I thought I had aced the test. I could hear everything, and I was just—thought that it was—just wonderful. And, uh, after the test—I came out of the room with a big smile on my face. Uh, he came into the office. He looked at the report and he said, 'Well—you do have hearing loss and it's bad enough that you do require to have hearing aids.' And then he looked at me and he said, 'It's about the same as my wife, uh, with her hearing loss and she refuses to wear hearing aids.' And he says, 'I know you won't wear 'em either.' And that was the—that's the end of the—that's the end of my story. He just said, 'I, I know at this stage of the game, you wouldn't wear the—you wouldn't wear the hearing aids.' (Carmen, Interview #16)

**Priority setting.**   For some participants, their narratives revealed that HA use was not a priority for them at the moment. Individuals who provided reasons for HA non-use around priority setting included narratives that described different health conditions or other family member's situations that required their immediate time, attention, or resources than HAs. For example, Anna stated, "So, I had the hearing test and she said I had some hearing loss and they

recommended hearing aids. Because I live in [a state] I called [the state university] and I made an appointment to go in. Uh, but shortly after coming home, I had a loss [death] in my family and I just let it [HA use] go." (Interview #13) Anna had a death in her family, while Karly had her own health condition that took priority over HA use:

Um, and then I put it [HA fitting/follow-up] off. I got diagnosed with ovarian cancer in, um, November of '16 so I had surgery and I had chemo. And that's been a while since I finished all of that. So, I mean chemo sounds like a really easy thing, but it leaves you with lots of issues. And, so, I have neuropathy in my feet. I have lymphedema from the surgery. I have, um, probably more hearing loss. And, um, so I did go [to the audiologist] and I waited 'til after I finished all that [cancer treatment] because I knew that that chemo could impact a hearing loss if you had it. (Anna, Interview #7)

## Discussion

In the present qualitative study, we gathered narratives and employed an attribution theory framework [25] to identify the reasons adults with HL, who were prescribed HAs, do not use them. Thematic analysis of 20 adults' narratives revealed the following nine themes for HA non-use. Internally motivated reasons included: (1) non-necessity, (2) stigmatization, (3) lack of integration into daily living, and (4) unreadiness due to lack of education. Externally motivated reasons of non-use included: (5) discomfort, (6) financial setback, (7) burden, (8) professional distrust, and (9) priority setting.

### Non-audiologic reasons for nonuse

The present study's unrestricted personal accounts from adults with HL who do not use their prescribed HAs revealed that there are a number of factors that impact HA nonuse and go beyond traditional, device-centric reasons (e.g., sound quality is poor, feedback, short battery life, etc.) [37]. Notably the reasons for HA nonuse that emerged from our participants' stories differed from those of Linssen et al [19], despite the fact that both data sets were evaluated in the context of the attribution theory framework. Although the studies uncovered reasons that are both internally and externally motivated, the present corpus revealed a variety of additional non-audiological reasons that did not emerge from Linssen et al's [19] semi-structured interviews. It is worth noting that some of the non-audiological reasons that emerged out of the present data set, hearken to reasons for HA non-used noted in Lockey et al [17]. However, their study differed in that the sample consisted of women only, who consistently wore their hearing aids. The Dutch study similarly revealed many internally motivated reasons that mirrored our present findings from the U.S. (e.g., denial of need and forgetting the HA), HA stigmatization did not emerge as an attribution for nonuse in their work. This is remarkable given that previous research [35, 37, 38] suggests stigmatization has long been a factor in HA uptake or the lack thereof, just as our adults' narratives revealed. Differences also arose across the two studies' externally motivated attributions for HA nonuse. In their study, Linssen and colleagues [19] found three external factors responsible for HA nonuse: (1) incompetence of the HA dispenser, (2) poor HA function, and (3) incompatible environment. Although valuable, when hearing healthcare providers focus on such traditional and device centric reasons for HA nonuse they run the risk of unknowingly removing the patient from the center of care, thus making it difficult to holistically evaluate a person's experiences with HAs during tasks of daily living [17, 29]. On the other hand, focusing on non-audiologic, externally motivated

reasons such as those we uncovered in the present study (e.g., professional distrust, priority setting, and financial setback) allow care providers to apply the ICF framework [29] and guide their person-centered care.

The aforementioned differences between the reasons for HA nonuse delineated in our findings and Linseen and colleagues' [19] could be accounted for by a number of reasons we touched on at the outset of this paper (e.g., the participants' countries of residence). It stands to reason that the variation in data collection across the studies—specifically, the narrative approach we employed in the present study—could be a major factor contributing to the different reasons for HA nonuse. The semi-structured interviews Linseen et al [19] utilized do not allow researchers to gather the same information attainable from a participant's unrestricted narrative. When researchers employed semi-structured interviews there is always the chance that the participants will adapt their answers to what the researcher wants to hear as a result of their questions or prompts (e.g., What about the HA prohibits you from using them?), thus resulting in filtered perspectives/insights. Because narratives are particularly suited as a means to communicate the human experience [39] not only does the narrative approach suit our exploration of reasons for HA nonuse, but it also naturally encourages/ allows for participants to share non-audiologic, person-centric factors that contribute to said nonuse. In the present study, allowing individuals who were diagnosed with HL and prescribed HAs to illuminate their reasons for HA nonuse via their personal narratives has contributed the most detailed and holistic understanding of HA uptake to date and one that can now drive necessary improvements to models of person-centered hearing healthcare.

## Implications for person-centered hearing healthcare

The present findings reiterate the importance of person-centered recommendations in the field of clinical audiology, particularly in regard to patients' treatment options. Person-centered care is an approach to healthcare that targets the needs of the patient and encourages them to be an active collaborator in their medical care [11]. When healthcare providers engage in this mutually beneficial relationship with patients (and their families), research shows that all parties involved benefit. For example, patients have better treatment follow-through and greater satisfaction with the care they receive when healthcare providers engage in person-centered care [40]. Thus, the diversity of narratives gathered in the present study—reflecting both internal and external attributions—remind us that without first placing the patient at the center of care and gauging their desire and willingness to try HAs, professionals may miss out on opportunities to engage in shared decision making with their patients. Shared decision making is a key aspect of person-centered care [11]. Without it, hearing healthcare providers might not only overlook the non-audiological factors that could contribute to an individual's HA nonuse, but the absence of person-centered care could result in professionals overlooking a wide variety of plausible, proactive treatment options that don't involve HAs but nonetheless facilitate communication and participation in daily living (e.g., personal sound amplification products (PSAPs) or communication strategy training).

The individuals who perceive HAs to be unnecessary in our narrative corpus, for example, could likely benefit from additional counseling by their care providers in the context of person-centered approach to managing their hearing healthcare. Hearing healthcare providers could utilize the ICF model [29] to guide their care and work closely with these patients to better understand their goals, in addition to their current functional abilities, daily activities, and most-frequent listening environments. The care provider and their patient could then use this information to drive decisions about their treatment options to improve both communication and participation in daily living. Subsequently, such person-centered care could result in

patients who give voice to this theme of *HAs are a non-necessity* becoming highly motivated to integrate HAs into their daily lives and note their holistic benefit.

Finally, for another example consider the attribution *HAs are a financial setback*. Financial commitment has long been considered a barrier in patients' decisions when considering the purchase of HAs in the U.S. [3, 41]—just as cost emerged as a reason for HA non-use in the present study. In the U.S., while the exact numbers may vary based on many different factors, including but not limited to bundled/unbundled costs and insurance coverage, patients can expect to pay on average $3,500 dollars for a set of digital HAs of a reasonable level of technology (not including costs of follow-up care and HA services performed by the hearing healthcare providers). For many individuals, especially those considered low-income, the amount of money that is required to obtain hearing healthcare services and treatment prevents them from committing to improving their hearing and overall health. Alternatively, when a hearing healthcare provider becomes aware that patient's HA nonuse is a result of financial concerns, they can turn to alternative solutions to support their patient and facilitate HA use, and later success. The 2017 U.S. Food and Drug Administration's approval of over-the-counter (OTC) HAs is a potential solution to the problem of financial commitment roadblocking patients' acceptance of HAs. While it may not be the ideal solution for every individual, those who are concerned with cost and cannot otherwise afford hearing healthcare will have a viable opportunity to improve their hearing, health, and communication [41]. In addition, better funding programs and financial assistance for hearing technology at both federal and state levels should be implemented for individuals across the lifespan. While some funding programs currently exist for both children and adults in the U.S. (e.g., Medicare, Medicaid, Starkey Hear Now Program, HIKE Fund Inc, etc.), many have income limitations and other criterion that continue to limit overall access for many middle-class families. Further adoption of more substantial financial assistance programs would improve hearing health of these individuals, which could lead to overall improved health in individuals and thus, lower healthcare costs across the board for all citizens at both state and federal levels.

### Limitations and future directions

While these present findings offer valuable insight into individuals' reasons for not using HAs, they are not without limitations. First, the demographics of the research participants lack diversity. Our study sample includes primarily White individuals, who are not representative of the hard-of-hearing community across the U.S. and elsewhere [1], thus it is difficult to generalize these data to the diverse populations that often seek hearing healthcare services across the world. Additionally, socioeconomic status was not noted during participant recruitment, so conclusions regarding potential associations between hearing healthcare access and financial well-being cannot be directly assessed. This is a limitation given the research that suggests a high prevalence of financial barriers within the hearing healthcare field [42], as well as what was found in the present study that financial concerns were a reason for individuals' HA non-use. The range in the current participants' ages should also be noted as a limitation. The youngest participant, at 27 years old, introduced a large age gap within the sample, with 15 of the remaining 20 participants reporting their age at or above 60 years. Lastly, the cause of HL was not included in our intake questionnaire. Acquiring this information could be beneficial as various causes of HL (i.e. congenital vs. presbycusis vs. sudden HL) could impact HA satisfaction and/or uptake for some individuals. Further qualitative research exploring the unrestricted personal accounts of individuals seeking hearing healthcare, while considering these limitations (e.g., by employing stratified random sampling), is necessary in order to gain an enhanced understanding of how we can provide a broader number of individuals with their ideal services.

## Conclusions

To conclude, the present study uncovered the following reasons that individuals with HL do not use HAs: (1) non-necessity, (2) stigmatization, (3) lack of integration into daily living, (4) unreadiness due to lack of education, (5) discomfort, (6) financial setback, (7) burden, (8) professional distrust, and (9) priority setting. These findings contribute to how professionals approach treatment provision for those individuals who choose not to utilize their HAs. Considering all barriers, even those that are non-audiologic in nature, may result in improved person-centered healthcare and patient satisfaction. Finally, identifying and understanding these internal and external attributions for HA nonuse is vital if professionals are to advance their efforts to increase HA uptake.

## Supporting information

**S1 Appendix. Interviewer's narrative interview script.**
(DOCX)

## Acknowledgments

The authors would like to acknowledge Kelsey Chandler for her help conducting interviews, in addition to Emma Brown and Nicole Trusty for their help transcribing the participants' narratives. We are also grateful to the members of the Aural Rehabilitation Lab at Utah State University for their help recruiting participants for the study. Finally, we would like to thank all of the individuals who were brave enough to share their stories with us about their hearing loss and challenges with hearing aids. This study would not be possible without them.

## Author Contributions

**Conceptualization:** Brittan A. Barker, Kristina M. Scharp.

**Data curation:** Caitlyn R. Ritter.

**Formal analysis:** Caitlyn R. Ritter, Brittan A. Barker.

**Methodology:** Kristina M. Scharp.

**Project administration:** Caitlyn R. Ritter, Brittan A. Barker.

**Resources:** Brittan A. Barker.

**Software:** Brittan A. Barker.

**Supervision:** Brittan A. Barker.

**Validation:** Caitlyn R. Ritter, Brittan A. Barker.

**Writing – original draft:** Caitlyn R. Ritter, Brittan A. Barker.

**Writing – review & editing:** Brittan A. Barker, Kristina M. Scharp.

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
