## [Decision Letter · Decision Letter 0]

2 Jul 2020

PONE-D-20-12039

Using attribution theory to explore the reasons adults with hearing loss do not use their hearing aids

PLOS ONE

Dear Dr. Barker,

Thank you for submitting your manuscript to PLOS ONE. After careful consideration, we feel that it has merit but does not fully meet PLOS ONE’s publication criteria as it currently stands. Therefore, we invite you to submit a revised version of the manuscript that addresses the points raised during the review process.

We look forward to receiving your revised manuscript.

Kind regards,

Sara Rubinelli

Academic Editor

PLOS ONE

Journal Requirements:

2. Please include additional information regarding the survey or questionnaire used in the study and ensure that you have provided sufficient details that others could replicate the analyses. For instance, if you developed a survey guide as part of this study and it is not under a copyright more restrictive than CC-BY, please include a copy, in both the original language and English, as Supporting Information

Additional Editor Comments (if provided):

Reviewers' comments:

Reviewer's Responses to Questions

**Comments to the Author**

1. Is the manuscript technically sound, and do the data support the conclusions?

Reviewer #1: Yes

Reviewer #2: Partly

2. Has the statistical analysis been performed appropriately and rigorously? 

Reviewer #1: N/A

Reviewer #2: Yes

3. Have the authors made all data underlying the findings in their manuscript fully available?

Reviewer #1: Yes

Reviewer #2: Yes

4. Is the manuscript presented in an intelligible fashion and written in standard English?

Reviewer #1: Yes

Reviewer #2: Yes

5. Review Comments to the Author

Reviewer #1: Thank you for the opportunity to review the manuscript titled “Using attribution theory to explore the reasons adults with hearing loss do not use their hearing aids”. The paper describes a qualitative study aimed at exploring reasons for hearing aid non-use in the United States. The topic is certainly relevant because of the increasing prevalence of hearing loss and, as clearly pointed out by the authors, because hearing aids have several benefits for individuals with hearing loss and it is thus important to identify ways to promote their use.

Overall the paper is well written and the authors did an excellent job at positioning their research within the broader context of research around hearing loss. Also, the aim of the study is clear and the methodology used (thematic analysis using an attribution theory framework) is sound. The discussion section nicely highlights the implications of the findings for person-centered hearing care.

I only have a few minor comments:

Methods

* I find it strange to see that one of the participants reported using a HA during all waking hours. Why was he not excluded? Even if he identified as a non-user, I think his perspective is different from the others.

* Phase II – III Details about software and tools (e.g., brand names) are not needed. E.g., the sentences “After data collection was complete, the interviewer uploaded the audio files from the digital recorder to an Apple iMac personal computer. Three research assistants experienced with narrative transcription then used Microsoft Word on an Apple iMac or Dell OptiPlex personal computer running Express Scribe Transcription Software Pro v 6.10 and Dragon Dictation v15 [31] paired with an INFINITY Foot Control IN-USB-2 transcription foot pedal and Sennheiser HD 280 Pro circumaural headphones to transcribe the recorded interviews” could be simplified to read as follows: “After data collection was complete, three research assistants experienced with narrative transcription transcribed the recorded interviews verbatim”

Results

* Please provide a brief definition of internally vs externally motivated reasons in the first paragraph.

* Table 2: I find the themes + description of the themes to be a bit redundant

Reviewer #2: 1.Marginally small sample (n = 20) for the narrative study design.

2. Too big age range (27 to 91 years).

3. Too big PTA range (15 to 50 dB, i.e. from normal hearing to moderate HL). Non-necessity COULD be a good non-usage reason for a person with close to normal (or even normal) hearing.

4. Only white people are included. Any reasons for it?

5. How could a person wearing HAs "during most waking hours" or even "during all waking hours" identify her(him)self as "non-user"? Can we trust other narrative information from such study participant? (3 persons with above-mentioned wearing time and 1 person wearing HAs during about a half of waking hours, which totals to 1/5 of study sample).

The authors realize these limitations. Almost all of them are listed in the Limitations and Future Directions section.

6. PLOS authors have the option to publish the peer review history of their article (what does this mean?). If published, this will include your full peer review and any attached files.

Reviewer #1: **Yes: **Nicola Diviani

Reviewer #2: No

---

## [Author Response · Author response to Decision Letter 0]

9 Jul 2020

Below you will find our responses to each reviewer’s comments. We were able to address all of the reviewers’ comments. We have also added an Appendix that includes our interview protocol. 

Response to Reviewer 1’s feedback

Thank you, Dr. Diviani, for your helpful feedback. 

1) I find it strange to see that one of the participants reported using a HA during all waking hours. Why was he not excluded? Even if he identified as a non-user, I think his perspective is different from the others. 

We understand and appreciate this perspective from both Reviewers 1 and 2 (see Reviewer 2’s comment (5)). We did not exclude this participant (or any others) based on their reported hearing aid (HA) use times because we are interested in the participants’ perceptions of themselves (and the stories they tell themselves and others) because research shows that human behaviors are shaped by our perceptions (for more detail see Moen, 2006, p. 63). Sometimes humans’ perceptions align with measurable quantitative data, sometimes they do not. For example, I could be a “healthy” weight according to the World Health Organization and measurements of Body Mass Index, but if I perceive I am “fat” I might change my eating behaviors and go on a diet. Or I could be 45 kilos overweight (“obese” according to the World Health Organization and measurements of Body Mass Index) but perceive that I am “healthy” and not engage in any behaviors that affect my weight. 

Thus, in the case of our study and current MS, if a participant thought of themselves as a HA non-user, then they are behaving (and making meaning in their life) accordingly. Their perception of who they are is what ultimately matters and their perception as a HA non-user actually matches the other participants’. We agree that it would be interesting to know how quantitative information, such as data logging, correlates with other qualitative variables (e.g., self-identity), however it is not standard procedure to include quantitative details about the sample in a qualitative narrative study such as ours. For example, reporting frequencies or other quantitative data erases the nuance of analysis (reducing it to a question of “how much” as opposed to a question of quality). We have revised our MS and clarified for the reader our decision-making process when it came to including all people who identify as “non-users” (p. 9 footnote). 

2) Details about software and tools (e.g., brand names) are not needed. 

Thank you for your feedback. We eliminated these details throughout our “Methods” sections.

3) Please provide a brief definition of internally vs externally motivated reasons in the first paragraph. 

We included a definition on p. 3.

4) Table 2: I find the themes + description of the themes to be a bit redundant. 

We agree with Reviewer 1 that many of the themes’ descriptions are redundant with their theme titles. However, for some remaining themes the descriptions add much-needed clarification that the themes’ labels cannot convey if they stand alone (e.g., stigmatization theme). Thus, we have decided to keep both the themes and descriptions in the Table for consistency/parallelism across the Table.

Response to Reviewer 2’s feedback

We are grateful for Reviewer 2’s comments. As they commented, “The authors realize these limitations. Almost all of them are listed in the Limitations and Future Directions section.” We have further addressed them below and expanded on them in the MS when possible (of particular note is Reviewer 2’s 5th question).

1) Marginally small sample (n = 20) for the narrative study design. 

When saturation is reached (i. e. no new identities emerge from the stories) no additional data points are required for analysis (Fusch & Ness, 2015). As noted in the “Verification procedures” section, saturation for our study was reached at Interview #11 (as determined retrospectively). We made an a priori decision to collect stories from 20 individuals based on the saturation points of other thematic narrative analyses conducted with samples of low-incidence clinical populations (e.g., Flood-Grady & Koenig Kellas, 2019). We also selected and N = 20 to give us enough stories to reasonably engage in referential adequacy.

2) Too big age range (27 to 91 years). Too big PTA range (15 to 50 dB, i.e. from normal hearing to moderate HL). 

Our study’s aim was to gather narratives from adults who did not use their prescribed HAs. Our final sample includes individuals all over the age 18-years-old, thus they fit the operational definition of adulthood. Furthermore, all of these individuals were diagnosed with permanent hearing loss and prescribed HAs by a hearing healthcare provider—our operational definition for hearing loss was not rooted in a particular degree or shape of hearing loss. Nonetheless, we recognize the experimental control that could be afforded by a study with a different design (and a more constrained participant eligibility criteria) than the one we used for the current study. 

3) Non-necessity COULD be a good non-usage reason for a person with close to normal (or even normal) hearing. 

Reviewer 2 is correct—in fact, all of the attributions that were illuminated in our study could be considered “good” reasons for HA nonuse. However, the aim of our study was to bring to light what reason HA non-users perceive as barriers to HA use. It was not our goal to assign value (or validity) to the reasons.

4) Only white people are included. Any reasons for it? 

Reviewer 2 is again correct. As we mentioned and discussed in our initial MS submission, our sample primarily consisted of White individuals, despite our public recruiting via social media, our lab’s website, etc. This was not intentional nor is it ideal, as we mention in the “Limitations and Future Directions” section of our paper. We elaborated more on why this can be a problem and how to potentially solve it in the future (p. 27). 

5) How could a person wearing HAs “during most waking hours” or even “during all waking hours” identify her(him)self as “non-user”? Can we trust other narrative information from such study participant? (3 persons with above-mentioned wearing time and 1 person wearing HAs during about a half of waking hours, which totals to 1/5 of study sample). 

We understand and appreciate this perspective from both Reviewers 1 and 2 (see our response to Reviewer 1’s comment (1) above for details). We did not exclude any participants based on their reported HA-use times because we are interested in the participants’ perceptions of themselves, not a quantitative measurement/assessment of whether or not they should be categorized as a non-user.

---

## [Decision Letter · Decision Letter 1]

18 Aug 2020

Using attribution theory to explore the reasons adults with hearing loss do not use their hearing aids

PONE-D-20-12039R1

Dear Dr. Barker,

We’re pleased to inform you that your manuscript has been judged scientifically suitable for publication and will be formally accepted for publication once it meets all outstanding technical requirements.

Kind regards,

Sara Rubinelli

Academic Editor

PLOS ONE

Additional Editor Comments (optional):

Reviewers' comments:

Reviewer's Responses to Questions

**Comments to the Author**

1. If the authors have adequately addressed your comments raised in a previous round of review and you feel that this manuscript is now acceptable for publication, you may indicate that here to bypass the “Comments to the Author” section, enter your conflict of interest statement in the “Confidential to Editor” section, and submit your "Accept" recommendation.

Reviewer #1: All comments have been addressed

Reviewer #2: All comments have been addressed

2. Is the manuscript technically sound, and do the data support the conclusions?

Reviewer #1: Yes

Reviewer #2: Yes

3. Has the statistical analysis been performed appropriately and rigorously? 

Reviewer #1: N/A

Reviewer #2: Yes

4. Have the authors made all data underlying the findings in their manuscript fully available?

Reviewer #1: No

Reviewer #2: Yes

5. Is the manuscript presented in an intelligible fashion and written in standard English?

Reviewer #1: Yes

Reviewer #2: Yes

6. Review Comments to the Author

Reviewer #1: All my and the other reviewer's previous comments have been addressed in a sufficiently detailed fashion and I have no further comments.

Reviewer #2: Authors made all requested correction and answered to all questions. The revised paper could be accepted

7. PLOS authors have the option to publish the peer review history of their article (what does this mean?). If published, this will include your full peer review and any attached files.

Reviewer #1: No

Reviewer #2: No

---

## [Editor Report · Acceptance letter]

27 Aug 2020

PONE-D-20-12039R1 

Using attribution theory to explore the reasons adults with hearing loss do not use their hearing aids 

Dear Dr. Barker:

I'm pleased to inform you that your manuscript has been deemed suitable for publication in PLOS ONE. Congratulations! Your manuscript is now with our production department. 

Kind regards, 

on behalf of

Dr. Sara Rubinelli 

Academic Editor

PLOS ONE